# Food Public Opinion Prevention and Control Model Based on Sentiment Analysis

**DOI:** 10.3390/foods13223697

**Published:** 2024-11-20

**Authors:** Leiyang Chen, Xiangzhen Peng, Liang Dong, Zhenyu Wang, Zhidong Shen, Xiaohui Cui

**Affiliations:** 1Key Laboratory of Aerospace Information Security and Trusted Computing, Ministry of Education, Wuhan 430001, China; cly_edu@whu.edu.cn (L.C.); pengxiangzhen2022@163.com (X.P.); dongliang0607@whu.edu.cn (L.D.); zhenyuwang@whu.edu.cn (Z.W.); shenzd@whu.edu.cn (Z.S.); 2School of Cyber Science and Engineering, Wuhan University, Wuhan 430001, China; 3Jiaxing Institute of Future Food, Jiaxing 314050, China

**Keywords:** food public opinion, food safety, SAE, personalized recommendations, sentiment prediction

## Abstract

Food public opinion is characterized by its low ignition point, high diffusibility, persistence, and strong negativity, which significantly impact food safety and consumer trust. This paper introduces the Food Public Opinion Prevention and Control (FPOPC) model driven by deep learning and personalized recommendation algorithms, rigorously tested and analyzed through experimentation. Initially, based on an analysis of food public opinion development, a comprehensive FPOPC framework addressing all stages of food public opinion was established. Subsequently, a sentiment prediction model for food news based on user comments was developed using a Stacked Autoencoder (SAE), enabling predictions about consumer sentiments toward food news. The sentiment values of the food news were then quantified, and improvements were made in allocating Pearson correlation coefficient weights, leading to the design of a collaborative filtering-based personalized food news recommendation mechanism. Furthermore, an enhanced Bloom filter integrated with HDFS technology devised a rapid recommendation mechanism for food public opinion. Finally, the designed FPOPC model and its associated mechanisms were validated through experimental verification and simulation analysis. The results demonstrate that the FPOPC model can accurately predict and control the development of food public opinion and the entire food supply chain, providing regulatory agencies with effective tools for managing food public sentiment.

## 1. Introduction

In recent years, food safety has increasingly drawn public attention, especially through social media, where incidents can rapidly escalate and significantly impact both the reputation of the food industry and public trust [1,2]. For instance, the E. coli outbreak linked to lettuce in North America sparked widespread health concerns, severely undermining consumer confidence in the safety of fresh produce. The swift spread of such incidents is driven not only by the gravity of the issues themselves, but also by the amplifying power of social media, which allows food safety concerns to intensify quickly, shaping strong and immediate public opinion.

Food public opinion refers to the diverse range of comments, viewpoints expressed, attitudes, information, and public opinion trends formed by the public and media regarding issues related to food safety [3,4,5]. Disseminated through social media, internet news, word-of-mouth, and other forms of information broadcasting, food public opinion can significantly impact public health risks, corporate reputation, and business operations, severely undermining consumer trust in the food industry [6]. According to the Ipsos public opinion big data monitoring system, in China, the volume of public opinion involving the catering industry alone in June was 46,711,736 pieces, with an average daily volume of 1,557,057 pieces. Strengthening the control of food public opinion can effectively reduce potential food safety risks and enhance consumer trust in the food industry [7].

To reduce the risks associated with food public opinion and ensure food safety, it is essential to strengthen the control of food public opinion and enhance its prevention from an informational perspective. In recent years, scholars both domestically and internationally have mainly focused on three aspects to strengthen the control of food public opinion risks. First, they explore enhancing the management of food public opinion by reducing information dissemination nodes, centralizing the release of food-related information online [8], and strengthening the promotion and training of food safety knowledge [9]. Second, they emphasize accelerating the construction of market and social regulatory measures, media guidance strategies, and risk communication strategies [10]. Third, they investigate the use of information technologies such as neural networks, machine learning, and natural language processing to monitor and provide early warnings for food public opinions [11,12]. Although these studies have somewhat improved the prevention of food public opinions, challenges remain due to the high sensitivity, high public involvement, and rapid spread of food-related public opinions. There are issues with the efficiency and precision of food public opinion prevention and control.

Deep learning technology excels in sentiment analysis, with deep learning models not only effectively identifying public sentiment towards food safety issues [13], but also recommending personalized sentiment information based on user preferences and browsing history. Zhang et al. developed the IFoodCloud system to systematically gather and analyze public sentiment on food safety across the Greater China region [14]. Through this system, we conducted an in-depth exploration of public perceptions of food safety, highlighting the potential of big data and machine learning in enhancing risk communication and supporting decision-making processes. Deep learning models can learn through self-training, which has led scholars both domestically and internationally to use deep learning for monitoring and early warning of food public opinion [14]. For example, Li et al. proposed an Att-BLSTM-CNN model [15] for opinion and sentiment analysis in the field of food safety, while Wang et al. developed a BERT-BLSTM-based model [16] for extracting entity relationships in food public opinion. Bouzembrak et al. developed a food fraud tool (MedISys-FF) for collecting [17], processing, and presenting food fraud reports published by media outlets worldwide. The collected reports were compared with other databases, demonstrating high relevance and aiding in understanding food fraud issues. Deep learning can provide intelligent means for the prevention and control of food public opinion [18,19,20]. This enables more accurate monitoring and control of food safety sentiment.

Personalized recommendation is a method that analyzes and predicts users’ interests and needs to provide them with personalized product or service recommendations [21,22,23]. Personalized recommendation technology uses data analysis and machine learning to predict user interests and needs based on their behavior, preferences, and demographic information. In the food industry, as food choices are deeply personal and influenced by health needs, lifestyle, and cultural factors, personalized recommendation plays a crucial role in the food industry by offering tailored food choices, dietary advice, and content that align with individual consumer preferences [24,25,26]. For instance, Wang et al. introduced a novel recommender system that enhancing the quality of food recommendations [27]. Gao et al. developed a specialized solution based on neural networks called Hierarchical Attention-based Food Recommendation (HAFR) [28]. In the context of food public opinion, by analyzing user behavior and sentiments, personalized news recommendations can be provided, thus offering a better means for managing food public opinion [29,30]. Additionally, personalized recommendations can support food safety and public health by providing timely and relevant information on food safety issues [31], recalls, and nutritional guidance [32,33] based on individual profiles.

This paper, starting from users’ sentiments towards food public opinion, combines deep learning and personalized recommendation algorithms to study a food public opinion prevention and control model based on sentiment analysis. First, a foundational framework for food public opinion prevention and control was constructed. Second, a food public opinion early warning technology based on user sentiment was designed using Stacked Autoencoders (SAEs) to enhance the “prevention” capability of food public opinion. Third, a user personalized recommendation algorithm based on sentiment analysis was designed using collaborative filtering and Bloom filters to improve the “control” capability of food public opinion. The food public opinion prevention and control model based on sentiment analysis can achieve integrated prevention and control of food public opinion and has high application value.

This paper makes several significant contributions to the research field, which are outlined as follows:(1)A foundational framework for food public opinion prevention and control was constructed.(2)A food public opinion early warning technology based on user sentiment was designed using stacked autoencoders (SAEs), enhancing the “prevention” capability of food public opinion.(3)A personalized recommendation algorithm based on sentiment analysis was designed using collaborative filtering and Bloom filters, enhancing the “control” capability of food public opinion.

The following sections of this paper are organized as follows. In Section 2, an overall design for the Food Public Opinion Prevention and Control (FPOPC) framework is presented. Section 3 focuses on the design of key methodologies, including user sentiment analysis related to food public opinion and a personalized recommendation mechanism. In Section 4, the implementation of the proposed model is detailed, along with a structural analysis. Finally, the paper concludes with a summary of the findings and offers a perspective on future research directions.

## 2. Materials and Methods

### 2.1. Food Public Opinion Analysis

The food supply chain is characterized by complex data flow, numerous circulation stages, large temporal and spatial gaps, and diverse stakeholders. We divide the entire supply chain into three parts: upstream, midstream, and downstream. The upstream includes the cultivation, storage, and initial processing of food-related raw materials, involving food raw material producers, initial processors, and transport personnel. The midstream encompasses food procurement, deep processing, finance, and marketing, primarily involving food processing plants, major suppliers, marketing personnel, and transport personnel. The downstream includes food retail, transportation, and consumption, involving distributors, retailers, and end consumers.

With the development of technologies such as the Internet and the Internet of Things, every stage of the food supply chain is now fully connected, and each stage is susceptible to the influence of online public opinion, which can significantly impact the entire food supply chain, as shown in Figure 1.

The development of food-related public sentiment can be divided into five stages: public opinion period, development period, peak period, fluctuation period, and decline period. The dynamics of food sentiment run through all stages of the food supply chain. During the public opinion period, online users begin reporting and sharing food-related issues, which then leads to the exposure of the food event. In the development period, the spread expands to a larger group, though a full-scale food-related public sentiment event has yet to form. The peak period is marked by reports and widespread coverage from official media and authoritative institutions, leading to a significant impact on the food supply chain. During the fluctuation period, stakeholders provide explanations and responses to the sentiment, causing fluctuations in public emotions. In the decline period, as time passes and related issues are resolved, public and media attention diminishes, and the sentiment is ultimately resolved.

We construct a sentiment analysis-based food sentiment prevention and control model, capable of real-time prediction and dynamic control of sentiment development and the various stages of the food supply chain, providing regulatory agencies with effective tools for controlling food-related public sentiment.

### 2.2. Food Public Opinion Prevention and Control Framework

Food public sentiment reflects consumer concerns and feedback on aspects such as food quality, additives, pesticide residues, labeling, and production processes. Strengthening the control of food public sentiment is a crucial measure to ensure food safety. We take consumer sentiment as the starting point, using deep learning models to analyze consumer comments on food-related public sentiment, predict their emotional tendencies, and identify key areas of concern. This provides strong support for managing public sentiment.

Additionally, we dynamically adjust the parameter weights of personalized recommendation algorithms based on model predictions to offer tailored food news recommendations to consumers. This helps food public sentiment management departments implement comprehensive monitoring and control. Figure 2 illustrates the food public sentiment prevention and control framework we propose.

We have designed a food public sentiment prevention and control framework aimed at serving sentiment management personnel and consumers. The framework consists of the following components: an underlying information collection/preprocessing module, a distributed storage database, a food sentiment early warning module, a food sentiment control module, and a top-level application module.

The information collection/preprocessing module is composed of a data collection unit and a preprocessing unit. The data collection unit is responsible for gathering user behavior data, including likes, favorites, number of views on food news, and detailed comments on various food-related news. The preprocessing unit handles the initial data processing, including data cleaning, standardization, and supplementation.

Data storage utilizes a distributed database, enabling data partitioning to improve system performance. The food sentiment early warning module employs several sentiment analysis models (SAEs) based on user comments to predict consumer sentiment towards food news and provide early warnings.

The food sentiment control module quantifies the sentiment prediction results into weights, optimizing the Pearson correlation coefficient weight distribution to offer personalized food news recommendations to users. Additionally, it incorporates an improved Bloom filter to enhance the performance of the recommendation system.

Finally, the top-level application module primarily supports sentiment management departments in providing comprehensive sentiment prevention and control, while also offering personalized food news recommendations to consumers.

### 2.3. Sentiment Analysis Methods

In our study, user behavior on food news is analyzed to understand public sentiment towards various food-related issues. Among different user interactions—such as browsing, liking, bookmarking, commenting, and sharing—comments are identified as the most reflective of users’ emotional reactions. Comments contain rich information about users’ opinions and feelings, making them an essential data source for analyzing public sentiment. We use a deep learning model, specifically Stacked Autoencoders (SAEs), to process user comments. This model analyzes the emotions conveyed by user comments regarding specific food-related public opinions. By doing so, we can predict user sentiments and achieve an early warning effect for food information public opinion.

#### 2.3.1. Stacked Autoencoders (SAEs)

SAE consists of multiple self-encoders (AEs) for stacking; each AE is a unit in the model used to create a deep network. A self-encoder is a type of artificial neural network primarily composed of an encoder and a decoder. It is commonly used in semi-supervised and unsupervised learning tasks. The main objective of a self-encoder is to learn a representation of the input data by using the input itself as the target during the learning process, effectively capturing and encoding essential features of the input information. Self-encoders learn the input *x* as a supervised signal and are also known as self-supervised learning algorithms. The purpose of autoencoder design is to learn the mapping relationship $f_{\theta }$ from input *x* to output $\bar{x}$. This process consists of two parts: the data encoding process and the data decoding process, namely the encoder and decoder. The data encoding process can be interpreted as reducing the high-dimensional input *x* into a low-dimensional latent variable *z*, represented by $g_{\theta 1}$. The data decoding process can be understood as decoding the input *z* into the high-dimensional *x*, represented by $h_{\theta 2}$. The design objective of the autoencoder is shown in Equation (Equation 1), the optimization process in Equation (Equation 2), and the optimization method in Equation (Equation 3).
(1)$$x\approx \bar{x}$$
(2)$$\bar{x}=h_{\theta 2}\left(g_{\theta 1}\left(x\right)\right)$$
(3)$$\;\mathrm{Minimize}\;\xi =\mathrm{dist}(x,\bar{x})$$

In the equations, the similarity between the input *x* and the output is measured using the Euclidean distance, as shown in Equation (Equation 4).
(4)$$\xi =\sum_{i}{\left(x-\bar{x}\right)}^{2}$$

In the process of reconstructing $\bar{x}$, if the number of hidden units is equal to or greater than the number of input units, it can affect the encoding process. Therefore, sparse constraints are introduced into the objective function to address this issue [28]. To enforce coefficient constraints *S*, we minimize both the reconstruction error $L(X,\bar{X})$ and Kullback–Leibler (KL) divergence $D_{KL}\left(p∥q_{j}\right)$. This is represented as Equation (Equation 5): (5)$$\begin{array}{cc} S & =L\left(X,\bar{X}\right)+\zeta \sum_{j=1}^{N_{h}}D_{KL}\left(p∥q_{j}\right)\\ \; & =\frac{1}{2}\sum_{i=1}^{n}{∥x^{i}-h_{\theta 2}\left(g_{\theta 1}\left(x\right)\right)∥}^{2}+\zeta \sum_{j=1}^{N_{h}}\left[p\log\frac{p\left(1-q_{j}\right)}{q_{j}\left(1-p\right)}+\log\frac{1-p}{1-q_{j}}\right] \end{array}$$

In the equation, $N_{h}$ denotes the number of hidden units, ξ represents the weight of the sparsity term, and *p* is a sparsity parameter close to zero. The average activation of hidden unit *j* over the training set is denoted as $q_{j}$, which equals $\frac{1}{N}\sum_{n=1}^{N}g_{\theta j}\left(x_{n}\right)$. If $p=q_{j},D_{KL}\left(p∥q_{j}\right)=0$, providing the sparsity constraint on the encoding.

The SAE model can be viewed as a stack of autoencoders, where each layer uses the output generated by the previous layer as its input, forming a deep network. The structure of SAE is depicted in Figure 3. SAE achieves efficient encoding and decoding of data by minimizing reconstruction errors through autoencoders. The training process is outlined in Algorithm 1.
**Algorithm 1** Training algorithm for SAE**Require:** Dateset $Data=\{x_{n}\}$;The number of hidden units $N_{h}$; Number of iterations: *m***Ensure:** Optimization results 1.Initialize the matrix and randomize the bias. 2.Construct $g_{\theta 1}\left(x_{n}\right)=g\left(U_{h}^{T}x_{n}+b\right)$ // In the encoding process, $U_{h}^{T}$ is the weight matrix between the input layer and the hidden layer, and *b* is the bias vector generated during encoding. 3.Perform forward propagation to reconstruct $\bar{x}$:$h_{\theta 2}\left(g_{\theta 1}\left(x_{n}\right)\right)=h\left(U_{r}^{T}g_{\theta 1}\left(x_{n}\right)+b^{\prime }\right)$ // Encoding process, where $U_{r}^{T}$ is the decoding matrix between the input layer and the output layer, and $b^{\prime }$ is the bias vector generated during decoding. 4.Calculate loss. 5.Perform backpropagation to update model parameters. 6.Repeat steps 2–5 *m* times to output the optimization results and use them as feature vectors of the input vectors to extract higher representations in SAE.

#### 2.3.2. Food Public Opinion Prediction Model

In the entire food supply chain, we believe that the prevention and control of the development of food-related public opinion mainly rely on the emotional feedback of netizens (consumers) regarding relevant food news or events. Therefore, we first analyze the relevant comments of consumers, categorizing them into positive and negative sentiments. We then design an SAE model to predict food public opinion periods in various stages of the food supply chain in advance, thereby providing a basis for the progression of related food news. The sentiment prediction model designed in this paper consists of an SAE, a predictor, and a weight quantifier. The model is illustrated in Figure 4.

The SAE model consists of a regression layer and multiple autoencoders, where the regression layer (output layer) is used to fit the output. After the model predicts food public opinion information and fits the output, the predictor is used to predict the emotional responses of users to relevant food public opinion news. The weight quantifier then quantifies the weights, providing support for personalized news recommendations.

The training of the SAE primarily involves a combination of backpropagation and gradient optimization. However, the food public opinion prediction model has high performance requirements for model training. Therefore, we refer to the method proposed in [29] to train the food public opinion prediction model.This paper mainly adopts a greedy layer-wise unsupervised learning algorithm. The core of this method is to perform layer-wise pre-training on the neural network and then use the backpropagation algorithm to adjust the model parameters, achieving better prediction results. The training process is illustrated in Algorithm 2.
**Algorithm 2** Sentiment prediction model training algorithm.**Require:** Dateset $Data=\{x_{n}\}$; The expected number of hidden layers *N*; The number of pre-training iterations is *T***Ensure:** Optimization results; Sentiment prediction; Weights 1:Collect early-stage user comments on food public sentiment, analyze them, calculate the number of negative and positive sentiments, and use them as the training dataset Data. 2:Collect user sentiment feedback on current food public sentiment within the initial time period *t*. 3:Set the weight of odd-numbered terms as ξ, Sparsity parameter *p*; Initialize the matrix and random bias vector. 4:Greedy layer-wise algorithm training for hidden layers. 5:Train the first layer using Algorithm 1. The input dataset is the training dataset Data. 6:Starting from the second layer, use the output of the hidden layer as the input for the next layer, obtaining the encoding-decoding matrix and bias vector for the next hidden layer. 7:Utilize backpropagation combined with gradient optimization techniques to perform top-down layered adjustments of parameters across the entire neural network. 8:Based on the data collected from step 2, make predictions to obtain overall positive and negative results. 9:Construct the weight quantifier: ϑ is the number of positive comments and ω is the number of negative commentsIf the prediction result is positive:Public sentiment index $\varepsilon =1+\frac{\vartheta }{\vartheta +\omega }$, increase recommendation weightIf the prediction result is negative:Public sentiment index $\varepsilon =1-\frac{\omega }{\vartheta +\omega }$, decrease recommendation weightIf the prediction result is neutral(ϑ=ω):Public sentiment index ε=1.

### 2.4. Personalized Recommendation Mechanism

Based on the prediction outcomes of the food public opinion model regarding user sentiment, we are able to provide timely warnings on the development of food-related public opinion at every stage of the food supply chain. Further, we improve the personalized recommendation mechanism to dynamically regulate the development of food-related public opinion. We predict users’ sentiment towards a certain food-related public opinion, and dynamically increase the proportion of news recommendation related to that food public opinion when the overall prediction is positive, and dynamically decrease the proportion of news recommendation related to that food public opinion when the overall prediction is negative. This is used for management to quickly intervene in food public opinion, reduce the heat of food public opinion, and minimize the scope and degree of influence of food public opinion on the entire food industry.

#### 2.4.1. Design of Personalized Recommendation Algorithm Based on Collaborative Filtering

In the recommender system, the recommendation algorithm initially applied is mostly a hotness algorithm, which indiscriminately pushes news and opinions of high public concern. Although this recommendation algorithm can ensure the high hotness of the recommended news to a certain extent, this method has the characteristic of “Thousands of people are alike”, which is not conducive to the regulation of public opinion. Personalized recommendation algorithms provide users with news of interest from their point of view, avoiding the waste of resources and reflecting the characteristic of “one person, one face”, which is gradually developing into the mainstream recommendation algorithms. Therefore, in this paper, we adopt the personalized recommendation algorithm based on collaborative filtering as a means to intervene in the development of food public opinion. Collaborative Filtering (CF) is a common recommendation algorithm, which analyzes users’ behaviors and preferences and discovers the similarities between users, so as to recommend items for users that they may be interested in. In this paper, food opinion prediction weights are added to the original recommendation algorithm to serve food opinion “control”. The conceptual diagram of the personalized recommendation algorithm based on collaborative filtering is presented in Figure 5.

This paper adopts a user-based collaborative filtering algorithm. The basic idea is to pre-calculate the similarity between news items based on the historical preference data of all users, and then recommend news similar to those liked by the user. The main process includes two parts. First, compute the set of news similar to the interests of the target user. Second, recommend news from this set that the target user has not yet viewed but is liked by similar users. In this paper, the Pearson correlation coefficient is used as the similarity measure. The independent ratings are adjusted by the user’s average rating to reduce the impact of user rating bias. The Pearson correlation coefficient calculation is demonstrated in Equation (Equation 6).
(6)$$\mathrm{sim}\left(u,v\right)=\frac{\sum_{i\in I}\left(r_{ui}-{\bar{r}}_{u}\right)\left(r_{vi}-{\bar{r}}_{v}\right)}{\sqrt{\sum_{i\in I}{\left(r_{ui}-{\bar{r}}_{u}\right)}^{2}}\sqrt{\sum_{i\in I}{\left(r_{vi}-{\bar{r}}_{v}\right)}^{2}}}$$
where $r_{ui}$ and $r_{vi}$ represent the degree of preference of users *u* and *v* for news item *i*, respectively. We quantify this value with a specific rating score to indicate the user’s preference for the food public sentiment news. The higher the value, the greater the degree of preference. ${\bar{r}}_{u}$ and ${\bar{r}}_{v}$ represent the average ratings of all news items interacted with by users *u* and *v*, respectively. By using the Pearson correlation coefficient, we obtain the similarity matrix of user *u*. Based on this similarity, we calculate the specific rating of user *u* for the target news item *t*, as shown in Equation (Equation 7).
(7)$$P_{u,t}={\bar{R}}_{u}+\frac{\sum_{k=1}^{2}S_{u,v}\left(R_{v,t}-{\bar{R}}_{v}\right)}{\sum_{k=1}^{2}S_{u,v}}$$
where $P_{u,t}$ is the final score of user *u* for news item *t*, ${\bar{R}}_{u}$ and ${\bar{R}}_{v}$ are the average scores of user *u* for other candidate news items, $S_{u,v}$ is the similarity (Pearson correlation coefficient) between users *v* and *u*, and $R_{v,t}$ is the specific rating of the news item *t* by the user *v*, who has the highest similarity to user *u*.

After calculating the ratings of the target user *u* for different news items using the above formula, compare the ratings (preference levels) for each news item. Then, combine this with the public sentiment index of the relevant food news to calculate the final rating weights and select the highest-rated items for recommendation. The personalized recommendation algorithm based on collaborative filtering is depicted in Algorithm 3.
**Algorithm 3** Collaborative filtering-based personalized recommendation algorithm**Require:** User historical behavior; Results of the food public opinion prediction model**Ensure:** Personalized recommendation list 1.Calculate the similarity matrix between user *u* and other users based on the historical ratings of all users for different news items. 2.Select the user vi most similar to user *u*, and based on the most similar user’s preference for the target news, predict and calculate the specific rating value of user *u* for the target news item. 3.Calculate the early warning index ϵ for all news items based on the sentiment prediction model, and compute the final user recommendation matrix. 4.Recommend news based on the rating values from highest to lowest.

#### 2.4.2. Design of a Fast Recommendation Mechanism Based on Bloom Filter

To ensure the rapid operation and security of the food public opinion control model, this paper adopts the distributed database HDFS and uses a Bloom filter to improve the retrieval process of the distributed database. HDFS features low data redundancy and high hardware fault tolerance. Files are stored in blocks with multiple replicas on the nodes of the cluster, ensuring hardware fault tolerance and preventing data loss in case of machine failure. Additionally, HDFS supports write-once, read-many operations, making it suitable for storing large-scale data files. The design of the fast recommendation mechanism based on the Bloom filter is shown in Figure 6.

A Bloom filter is essentially a very long binary vector and a series of random mapping functions. It is commonly used to quickly check whether an element belongs to a set. In our scenario, we add a corresponding Bloom filter to each cluster node in HDFS, and pre-map the data shards to the Bloom filters. This Bloom filter is used to quickly determine the location of relevant data, facilitating fast recommendations for food news. Here is the specific process for writing news data:Step 1.The client sends a write request to the NameNode.Step 2.Verify file existence and permissions quickly using a Bloom filter. Upon successful verification, the operation is logged directly to the EditLog before returning the output stream object.Step 3.The client partitions the file into 128 MB blocks.Step 4.The client sends the allocated writable DataNode list returned by the NameNode along with the data to the nearest first DataNode. Subsequently, the client and the multiple DataNodes allocated by the NameNode form a pipeline. The client writes data to the output stream object.Step 5.After writing each block, each DataNode returns a confirmation message.Step 6.Close the output stream after completing the data write.Step 7.Send a completion signal to the NameNode.

The specific process for news data retrieval (read) is as follows:Step 1.The client accesses the NameNode to query metadata information and obtain the list of data block locations for the file, and then returns the input stream object.Step 2.Select the nearest DataNode server and request to establish an output stream.Step 3.The DataNode reads data into the output stream and verifies it in packets.Step 4.Close the output stream.

### 2.5. Datasets and Experimental Configuration

We selected the IMDB and Amazon datasets based on key factors aligned with our research objectives:(1)Diversity of content and domain: The IMDB dataset covers movie reviews, while the Amazon dataset includes product reviews from various categories. This diversity allows us to evaluate the robustness and generalizability of our sentiment analysis model across different types of user-generated content.(2)Dataset size and balance: Both datasets are large, containing tens to hundreds of thousands of samples, ensuring our models are trained on comprehensive, representative samples of user sentiment. Additionally, the IMDB dataset is balanced with an equal number of positive and negative reviews, making it ideal for binary classification and performance comparison.(3)Clear sentiment labels: both datasets offer explicit sentiment labels (positive and negative), which simplifies model evaluation by providing clear benchmarks for classification accuracy.(4)Availability and accessibility: these datasets are widely used in sentiment analysis research and are readily accessible to the research community, promoting reproducibility and enabling comparative studies.

A brief description of the selected datasets is provided below.

IMDB [34] contains 50,000 user reviews, which are categorized into two classes: negative and positive. Reviews with an IMDB rating of less than 5 are labeled as 0 (negative, neg), while reviews with a rating of 7 or higher are labeled as 1 (positive, pos). The dataset is split into 25,000 reviews for the training set and another 25,000 for the test set. Both the training and test sets are stored in separate folders, with each folder containing both positive (pos) and negative (neg) reviews.

Amazon [35] spans over 10 years and contains approximately 500,000 reviews up to October 2012. The reviews include product information, user details, ratings, and full-text comments. It also encompasses reviews from all other categories on Amazon.

All experiments were conducted on our laboratory server, which is equipped with the following specifications: Windows Server 2022 Standard as the operating system, a 14th generation Intel(R) Core(TM) i9-14900K processor clocked at 6.0 GHz, 128 GB of RAM, and an GPU-NVTHGX-A100-SXM4-88D graphics card. The experiments were performed using Python 3.8.6 as the software environment.

## 3. Results and Analysis

### 3.1. Comparative Analysis of Food Public Sentiment Prediction Models

In this paper, we compare it with different deep learning models, and we can see that, compared with the LSTM model and the GRU model, the model designed in this paper has the best performance in terms of the MAE and RMSE metrics, especially in positive sentiment classification, with a smaller error and higher stability. Metrics results are presented in Table 1, Table 2, Table 3 and Table 4.

Figure 7 presents a comparison of MA and RMSE for the LSTM, GRU, and SAEs models in the context of Positive sentiment. LSTM Model: The MAE of the LSTM model fluctuates between 37 and 44. The lowest MAE is observed in model (a), while the RMSE reaches its peak in model (b), indicating a higher error in this configuration. The RMSE shows relatively larger fluctuations, suggesting that the LSTM model may struggle to consistently handle positive sentiment classification in certain cases.

GRU Model: The GRU model shows stable performance in MAE, ranging from 42 to 44 across the different models. Notably, model (g) achieves the lowest MAE. While the overall RMSE for GRU remains high, the fluctuations are relatively minor, indicating a degree of stability in its performance.

SAEs Model: The SAEs model performs exceptionally well in terms of MAE, maintaining low error values across nearly all models, with models (a) and (g) achieving the lowest MAE. The RMSE for SAEs is also relatively stable, indicating that the model not only excels in MAE, but also exhibits strong overall error control.

Figure 8 presents a comparison of MAE and RMSE for the LSTM, GRU, and SAEs models in the context of negative sentiment:

LSTM Model: The MAE and RMSE of the LSTM model exhibit relatively large fluctuations across all models. The MAE decreases in models (f) and (g), but overall remains at a high level, ranging from 42 to 44. The RMSE reaches a lower value in model (f), but is relatively high in other cases, peaking at 54.36 in model (e), indicating unstable performance.

GRU Model: The MAE of the GRU model remains relatively stable, consistently around 42, with model (g) performing the best and showing the smallest error. The RMSE does not vary significantly, but reaches its highest value of 54.29 in model (e), suggesting that the error distribution in negative sentiment classification is more uniform.

SAEs Model: The SAEs model achieves the lowest MAE, with minimal fluctuations, displaying the smallest errors among all models, particularly in models (a) and (g). This demonstrates its superior performance in negative sentiment classification. The RMSE is also relatively low, consistently ranging between 45 and 46, indicating good overall error control and strong model stability.

### 3.2. Analysis of Recommendation Efficiency

To better evaluate the proposed algorithm model, this paper calculates the recall rate to assess the comprehensiveness of the model’s recommendations. The recall rate is the ratio of the total number of items that the model recommends and the user has actually interacted with, to the total number of items the user has interacted with. In the equation, *R(u)* denotes the *N* items recommended to user *u*, which *T(u)* represents the set of items that user u liked in the test set. The recall rate is defined in Equation (Equation 8).
(8)$$Recall=\frac{\sum_{u}\left|R\left(u\right)\cap T\left(u\right)\right|}{\sum_{u}\left|T\left(u\right)\right|}$$

Figure 9 illustrates the changes in recall rate of the designed recommendation model as the number of iterations increases.

We used Recall, a commonly used performance metric in recommendation systems, as the primary evaluation method. By calculating the Recall@k values for different users, we can assess the performance of the recommendation system after sentiment analysis. If the model achieves a high Recall@k value, it indicates that our sentiment analysis effectively helps users find food items they might like, thereby enhancing the user experience.

Additionally, we can further analyze the performance of Recall@k across different time periods, user groups, or product types to optimize our recommendation strategy. This detailed analysis allows us to continuously improve the model, ensuring that the recommendation system better aligns with users’ actual needs.

## 4. Discussion and Conclusions

Sentiment analysis has emerged as a critical tool for gauging public attitudes towards food safety, product quality, and related issues. By analyzing sentiment data from social media, review platforms, and other channels, researchers can identify potential risks in public opinion, providing timely early warnings to food companies and regulatory bodies. Deep learning models, particularly Stacked Autoencoders (SAEs), have demonstrated remarkable effectiveness in handling large-scale text data, accurately extracting sentiment features, and offering valuable insights. In applications for monitoring food-related public opinion, personalized recommendation systems leverage sentiment analysis results to predict public behavior, enabling targeted interventions and enhancing precision in response efforts.

To enhance the precision of food public opinion prevention and control, and to strengthen the regulatory capabilities of management departments, this paper combines deep learning algorithms with recommendation systems to propose a sentiment analysis-based food public opinion prevention and control model. First, we conducted an in-depth analysis of the food public opinion development process, providing a theoretical foundation for the model. Next, based on an analysis of the stages of food public opinion development, we utilized a Stacked Autoencoder (SAE) to design a sentiment-based public opinion early warning system, which can predict user sentiment responses in advance. Following that, by integrating Bloom filters and distributed databases with the food public opinion prediction results, we developed a sentiment analysis-based personalized recommendation algorithm to achieve precise control of food public opinion. The final results demonstrate that the designed model can effectively and accurately control food public opinion. This study offers a practical and feasible solution for accelerating the digital transformation of the food industry, enhancing public opinion regulation capabilities, and ensuring food safety.

### 4.1. Limitations

The main limitation of this research is the reliance on a single modality that uses textual data as the primary source for sentiment analysis and opinion monitoring. In today’s Internet environment, opinion data are increasingly multimodal, including not only text, but also images, video, and audio, especially user-generated content on social platforms. These multimodal data types can provide richer insights into public sentiment and opinion, capturing nuanced responses that may not be fully conveyed by text alone. Our study is based on unimodal text data, which yields results that are generalizable, but may lack the specificity that multimodal analysis can provide. Therefore, future research should multimodal data to gain deeper insights and address this limitation.

Although the Food Public Opinion Prevention and Control (FPOPC) model is tailored for food-related public opinion, food-related issues have unique characteristics such as high negativity, fast spreading and long duration, and the model may face challenges in applying to a wider range of areas other than food public opinion.

### 4.2. Suggestions and Further Research

In the current internet environment, food safety issues are frequent, and the risks of food-related public opinion are pervasive. It is recommended to further encourage industry enterprises to actively engage in food safety education to enhance public awareness. At the same time, government agencies should adopt cutting-edge technologies such as digital humans and large models to intelligently improve the efficiency of public opinion guidance and management. Additionally, efforts should be made to strengthen the guidance, oversight, and regulation of self-media and social platforms, preventing the spread of misleading reports on food safety risks.

The findings of this study provide valuable insights into the application of advanced technologies in the field of food security in China, and offer clear direction and guidance for the development of future food public opinion prevention and control mechanisms.

In the next phase of research, we will further enhance the multimodal attributes of data sources such as images, videos, and audios to capture a wider range of public sentiment, integrate multimodal data for comprehensive analysis, and improve the model’s ability to perceive the overall dynamics of food-related public opinion.

Through transfer learning or fine-tuning methods, the FPOPC model can be extended to domains other than food public opinion, such as healthcare or environmental protection. Such tuning will increase the flexibility and generalizability of the model across different industries and public concern contexts.

## Figures and Tables

**Figure 1 foods-13-03697-f001:**
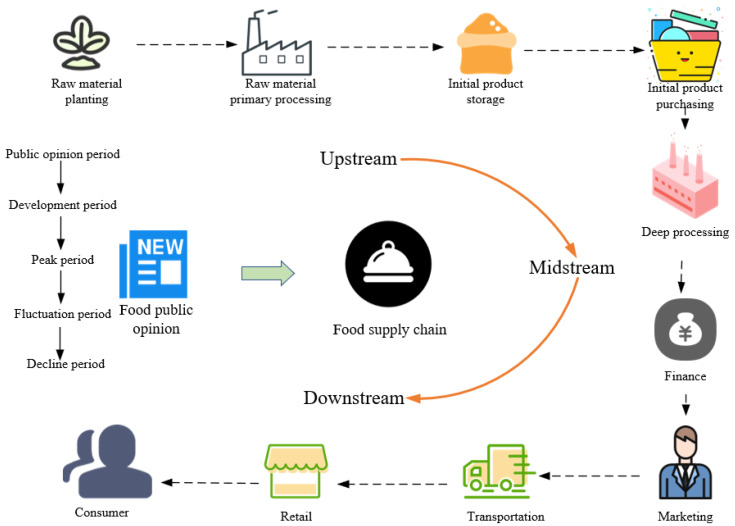
Diagram of food public sentiment development.

**Figure 2 foods-13-03697-f002:**
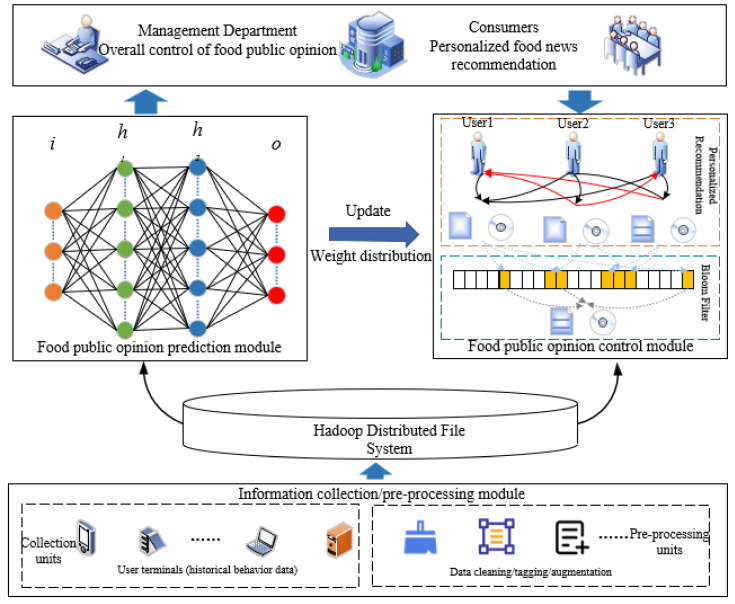
Diagram of the food public sentiment prevention and control framework.

**Figure 3 foods-13-03697-f003:**
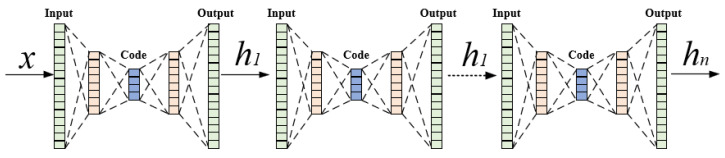
The structure of SAE.

**Figure 4 foods-13-03697-f004:**
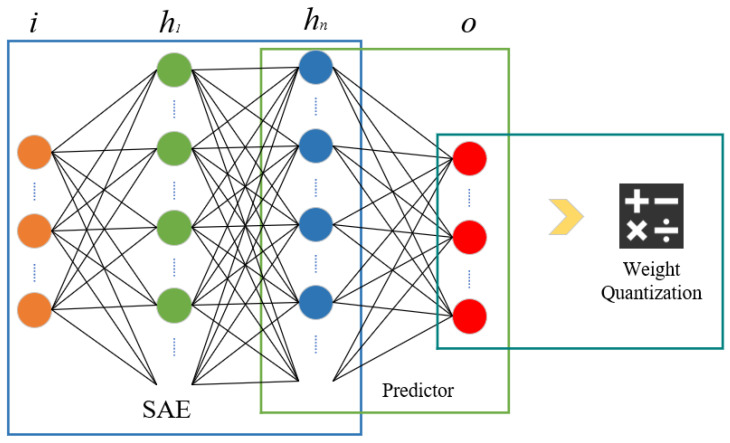
Structure of the food public opinion prediction model.

**Figure 5 foods-13-03697-f005:**
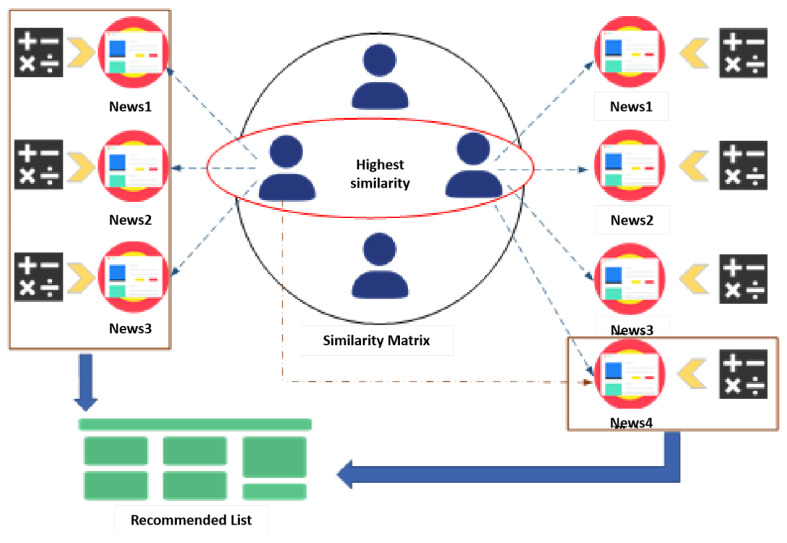
Diagram of the personalized recommendation algorithm based on collaborative filtering.

**Figure 6 foods-13-03697-f006:**
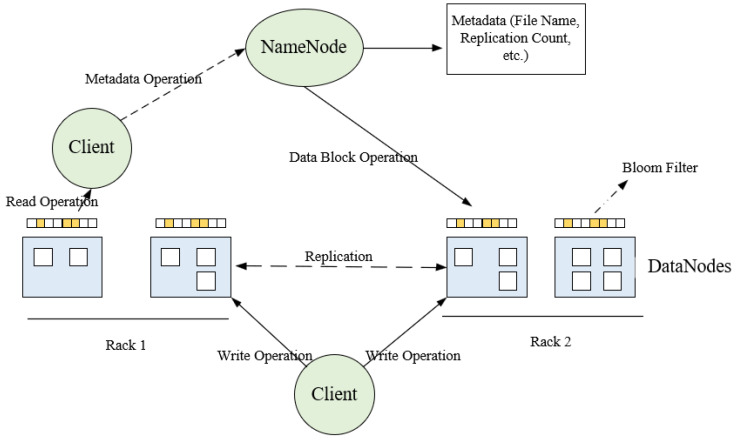
Fast recommendation mechanism based on Bloom filter.

**Figure 7 foods-13-03697-f007:**
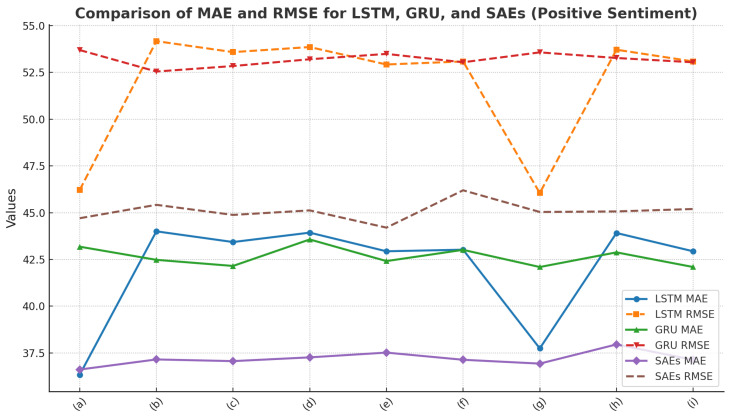
Comparison of MAE and RMSE for LSTM, GRU, and SAEs (Positive Sentiment).

**Figure 8 foods-13-03697-f008:**
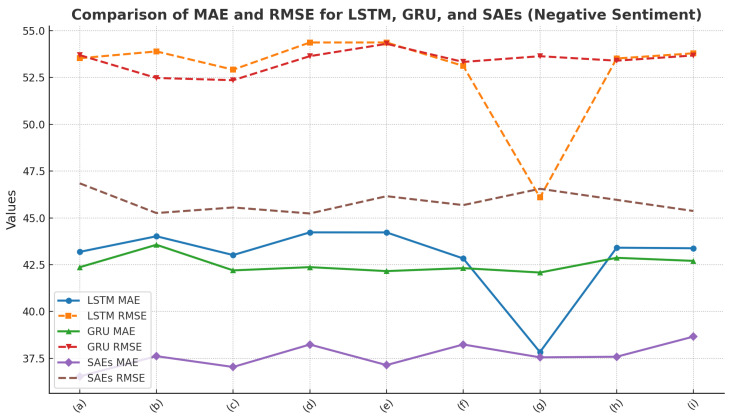
Comparison of MAE and RMSE for LSTM, GRU, and SAEs (Negative Sentiment).

**Figure 9 foods-13-03697-f009:**
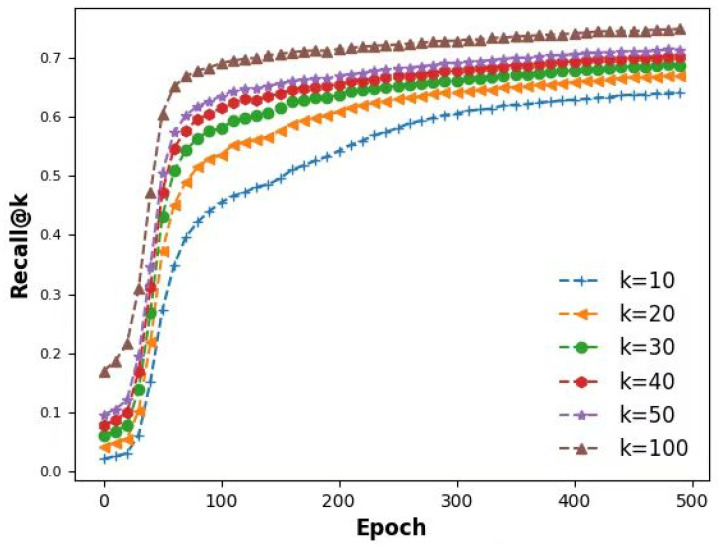
Recall rate of the designed recommendation model.

**Table 1 foods-13-03697-t001:** Metric results of different models in positive sentiment analysis in MAE.

	LSTM_MAE_Positive	GRU_MAE_Positive	SAEs_MAE_Positive
a	36.3355	43.1805	36.6117
b	44.0014	42.4770	37.1512
c	43.4291	42.1506	37.0612
d	43.9312	43.5672	37.2626
e	42.9348	42.4110	37.5173
f	43.0195	43.0111	37.1335
g	37.7423	42.0936	36.9283
h	43.9128	42.8770	37.9532
i	42.9348	42.0936	37.1335
AVE	42.1228	42.6407	37.1947

**Table 2 foods-13-03697-t002:** Metric results of different models in positive sentiment analysis in RMSE.

	LSTM_RMSE_Positive	GRU_RMSE_Positive	SAEs_RMSE_Positive
a	46.2119	53.6955	44.7031
b	54.1728	52.5473	45.4192
c	53.5847	52.8406	44.8794
d	53.8558	53.2033	45.1218
e	52.9233	53.4887	44.1996
f	53.0860	53.0416	46.1958
g	46.0506	53.5712	45.0367
h	53.7135	53.2707	45.0661
i	53.0860	53.0416	45.1958
AVE	51.9217	53.1920	45.0908

**Table 3 foods-13-03697-t003:** Metric results of different models in negative sentiment analysis in MAE.

	LSTM_MAE_Negative	GRU_MAE_Negative	SAEs_MAE_Negative
a	43.1847	42.3658	36.5362
b	44.0187	43.5638	37.6147
c	43.0146	42.1956	37.0368
d	44.2266	42.3691	38.2365
e	44.2237	42.1576	37.1389
f	42.8318	42.3152	38.2387
g	37.8326	42.0816	37.5517
h	43.4075	42.8674	37.5836
i	43.3780	42.7036	38.6582
AVE	42.2167	42.5133	37.6217

**Table 4 foods-13-03697-t004:** Metric results of different models in negative sentiment analysis in RMSE.

	LSTM_RMSE_Negative	GRU_RMSE_Negative	SAEs_RMSE_Negative
a	53.5260	53.6943	46.8457
b	53.8911	52.4725	45.2576
c	52.9147	52.3528	45.5567
d	54.3683	53.6385	45.2352
e	54.3683	54.2958	46.1586
f	53.1168	53.3287	45.6825
g	46.0900	53.6351	46.5538
h	53.5133	53.3957	45.9635
i	53.7897	53.6741	45.3685
AVE	51.8864	53.3875	45.8469

## Data Availability

The original contributions presented in the study are included in the article, further inquiries can be directed to the corresponding author.

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
