# Peer review of "Food Public Opinion Prevention and Control Model Based on Sentiment Analysis"

_foods, 2024, doi:10.3390/foods13223697_

Round 1
Reviewer 1 Report
Comments and Suggestions for Authors
The research subject is very interesting. In the Abstract, the authors could better explain the results and effective contributions of the research. The content of Figures 1 (page 4), 2 (page 5), 4 (page 8) and 6 (page 12) could be in English to facilitate readers' understanding. In section "2.3. Sentiment Analysis Methods" (page 6), it would be important to explain the aforementioned method in greater detail. In section "3.1. Datasets and Experimental Configuration", it would be appropriate to make clearer what the selection criteria were for the datasets considered in the research. The results, in general, are presented adequately. However, section "4. Discussion and Conclusions" is a weak point of the article. It lacks a better approach to the theoretical implications, making a counterpoint with the accessed literature (or previous studies), and the implications of the research. The limitations of the research and suggestions for future studies could also be increased. Respectfully, these are my comments and suggestions to the authors.
Reviewer 2 Report
Comments and Suggestions for Authors
Dear Authors,
The manuscript entitled “Food Public Opinion Prevention and Control Model Based on Sentiment Analysis” deals with an interesting and current topic. It has, however, some issues to be handled.
The term “food public opinion” sounds strange; it is much better to use “public opinion on food.” Personalized recommendation is not a technology; and is defined with itself in lines 64-66. The sentence in lines 70-71 does not fit in the logical flow of the text, since it has already been discussed that personalized recommendation has begun to be applied in the food industry. The presentation of existing literature related to the topic is too short; the significance of the topic, and the previous research results on it should be detailed here. Subsection 2.1 rather belongs to the previous section. Here, “transport personnel” is mentioned in two supply chain parts, moreover, transportation is said to be part of the third supply chain part. Several figures are incomprehensible, since there are a lot of Chinese words on them. The term “sentiment fermentation” is strange. The source of the five stages of the development of food-related public sentiment is missing. Subsection 3.1 rather belongs to the previous section. Explanation of Eq (8) variables are missing (or some of them are explained later).
Comments on the Quality of English Language
There are several spelling and grammar mistakes, especially the unnecessary use or missing spaces, see, e.g., lines 46, 49, 52, 62, 68, 70, 116, 129, 175, 188, 193, 194, 218, 221, 223, 278, 333, 337, 338, 352, 358, 405, and Algorithms 1 and 2.
Reviewer 3 Report
Comments and Suggestions for Authors
Dear Authors,
Food security is one of the biggest contemporary issues, therefore I highly appreciate the idea and your entire paper. Though, please present in the Introduction section some significant exemples to explain the relevance of the subject.
Between lines 58-60 you mentioned the concept of food fraud. Please explain its content in the context of your research.
In the paragraph between lines 64-73 you describe the personalized recommendation technology. Please, be more explicit about its content, importance, and multiple uses in the food industry. Also, mention the benefits of this technology for the food market, including in this context all interested stakeholders.
Before presenting the contributions of your work in the field, it would be valuable to identify previous contributions in the field and which are the differentiations that you propose.
Please translate into English all the figures included in the text.
Between lines 117-128 you describe the stages of food-related public sentiment. If these are your own ideas, there is nothing to be changed. But if you have been inspired by the literature that you have read, it is necessary to mention all the references.
Conclusions are elusive. It is unclear which are the practical outcomes for consumers and also for all companies involved.
In my opinion, the major weakness of the article is the lack to present the results of other studies on the same issue, and which are the differences proposed by the authors. If the methodology is completly new for the food industry, it would be appropriate to mention other domains in which similar technologies are used and their main outcomes.
Round 2
Reviewer 1 Report
Comments and Suggestions for Authors
Considering the adjustments or improvements made to the article, reported in the document "author_response.pdf", I understand that they contemplate the recommendations made. I congratulate the authors for the research and the effort in qualifying the article presented.
Reviewer 3 Report
Comments and Suggestions for Authors
I would like to congratulate authors, once again, for their work.
In the present form, I consider that the article meets the criteria for publishing.